# Pitfalls in diagnosing temperature extremes

**Lukas Brunner [1] ✉ & Aiko Voigt [1]**

Worsening temperature extremes are among the most severe impacts of human-induced climate change. These extremes are often defined as rare events that exceed a specific percentile threshold within the distribution of daily maximum temperature. The percentile-based approach is chosen to follow regional and seasonal temperature variations so that extremes can occur globally and in all seasons, and frequently uses a running seasonal window to increase the sample size for the threshold calculation. Here, we show that running seasonal windows as used in many studies in recent years introduce a time-, region-, and dataset-depended bias that can lead to a striking underestimation of the expected extreme frequency. We reveal that this bias arises from artificially mixing the mean seasonal cycle into the extreme threshold and propose a simple solution that essentially eliminates it. We then use the corrected extreme frequency as reference to show that the bias also leads to an overestimation of future heatwave changes by as much as 30% in some regions. Based on these results we stress that running seasonal windows should not be used without correction for estimating extremes and their impacts.

Percentile-based temperature extremes are defined as events of a certain rarity and are frequently used to derive downstream impact metrics such as heatwave properties, particularly in a warming climate. The aim of percentile-based extreme definitions is to account for spatial [e.g.,[1,2]] and seasonal [e.g.,[3–5]] variations in temperature distributions, so that extremes can occur across the globe and throughout the year. In addition, percentile-based extremes are intended to offset differences between datasets such as models, reanalyses, and observations [e.g., [2,6]].

The rarity of the extreme is set by calculating exceedances of an appropriate percentile threshold, typically on a daily basis. For example, on average 10% of days are expected to have a maximum temperature exceeding the 90th percentile when evaluated in the same period as the threshold is calculated. This base period is frequently chosen as 1961–1990 as recommended by the Expert Team on Climate Change Detection and Indices (ETCCDI; http://etccdi.pacificclimate.org/list_27_indices.shtml;[7]).

The period length of 30 years is a compromise between the length of the time series on the one hand and the number of samples available for the percentile calculation on the other hand. In particular observational datasets often do not provide high-quality data for long time periods, and longer periods may contain strong forced warming,

which is often undesirable. Short time periods, in turn, limit the number of samples on which the percentile threshold is based, which can lead to considerable biases[8]. Therefore, individual studies have also used longer periods[5,9–12].

The ETCCDI recommends a 5-day running window across the seasonal cycle to counter the limited number of samples. Combined with the recommended 30-year period, this results in a sample size of 150 values per calendar day for the percentile calculation. However, many recent heatwave studies do not follow this recommendation and use longer windows of 15 days[4,5,9,10,12–19] or even 31 days[6,20–28]. Notably, the widely used Heat Wave Magnitude Index daily (HWMId) is defined based on a 31-day running window[29]. The decision to deviate from the ETCCDI recommendation and use a longer window is often not explicitly motivated in the literature. However, it can be assumed that the intention is to increase the sample size beyond 150 values in order to avoid biases in the frequency of exceedances[8] as well as large variations in the extreme threshold between adjacent calendar days[16].

In the following, we reveal serious pitfalls when using such long-running windows. We show that they lead to large and systematic biases in the frequency of temperature extremes and discuss how these biases undermine commonly accepted assumptions about the properties of percentile-based extreme definitions. We demonstrate a

[1]Department of Meteorology and Geophysics, University of Vienna, Vienna, Austria. ✉e-mail: l.brunner@univie.ac.at

simple solution that essentially eliminates the bias and uses the corrected frequencies as a reference to investigate the effect of the bias on estimates of extreme changes under warming.

## Results

### Systematic biases in percentile-based temperature extremes

We define a bias in the frequency of daily maximum temperature extremes as a deviation from the theoretically expected extreme frequency. As an example: for the 90th percentile the expected frequency is 10% extreme days on average when calculating threshold and extremes in the same period (see methods for details). However, we find that in the ERA5 dataset, the 1961–1990 average, global average, daily maximum temperature extreme frequency based on the 90th percentile using a 31-day running window (TX90p31w) is only 9%, a relative bias of −10%. Regionally the bias can be considerably larger and exceed −30% as shown in Fig. 1a for ERA5 and in Fig. S1 in the supplementary information for the CMIP6 multi-model mean.

In individual months, the bias can be even larger, exceeding −75%. This almost excludes the occurrence of extremes altogether as shown in Fig. 1c for the example of a grid cell with a strong bias in the North Atlantic. This striking underestimation of threshold exceedances, particularly in the transition seasons, may seem counter-intuitive: The moving window used in the percentile calculation is symmetric around each calendar day—it encompasses both seasonally colder as well as seasonally warmer values. One might expect these to average out and lead to the desired smooth and well-defined extreme threshold, which is exceeded for ~10% of days throughout the seasonal cycle and across the globe. But this expectation does not hold as the threshold is, by design, an extreme percentile and not an average. It is, therefore, dominated by the seasonally warmer days in the window, making it exceedingly unlikely for the considered central day to exceed the threshold if (1) the seasonal gradient is strong and (2) the day-to-day variability is low.

The strongest bias is, hence, found in regions and seasons with a strong seasonal gradient but weak day-to-day variability, as showcased in Fig. 1c and Fig. S2. While many of the strongest biases, therefore, occur over oceans there are also several land regions with considerable bias such as India and the western US (Fig. S2a, c). Across most of Europe the strong seasonal cycle is offset by strong day-to-day variability, and only a weak bias remains (Fig. S2e). In regions with weak seasonal variations, such as most of the tropics and the Southern Ocean (Fig. S2i), the bias is consequently negligible. In the following, we will refer to this bias as running window bias.

To further investigate and contextualise this running window bias, we build on a study by Zhang et al.[8] based on synthetic data. As detailed in the methods, we include an additional seasonal cycle of varying amplitude into their auto-correlated white noise data. Figure 2a shows the frequency bias in absence of a seasonal cycle reproducing their results. For the in-base case, the same random sample is used to calculate the threshold and the exceedances; for the out-of-base case, exceedances are based on but the same threshold but a new random sample. The bias for the 5-day window is equivalent to Zhang et al.[8], while we use a slightly different second window size of 31 days (as opposed to their 25 days) to be consistent with the rest of our work.

The difference between the out-of-base and in-base and bias is shown in Fig. 2b. The temporal inhomogeneity introduced by this difference is widely recognised in the community and can be corrected following the recommendation by the ETCCDI and Zhang et al.[8]. For the 5 day window, the bias (Fig. 2a) and the difference (Fig. 2b) grow strongly with increasing percentile and vary periodically due to the the limited number of samples, as discussed in detail by Zhang et al.[8]. Based on this result alone, it is understandable why a longer window seems preferable, with the bias in Fig. 2a,b remaining much lower for the 31-day running window, especially for high percentiles.

### (a) Extreme frequency bias (%)

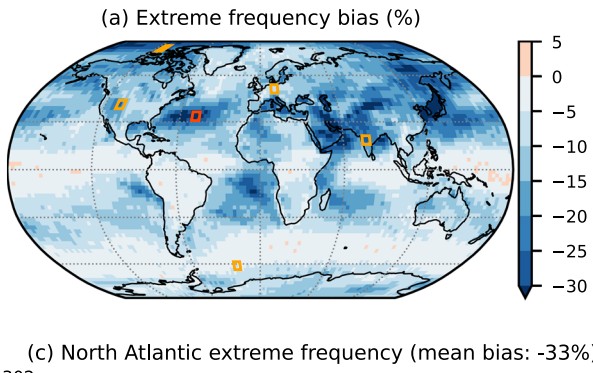

### (b) Corrected: Extreme frequency bias (%)

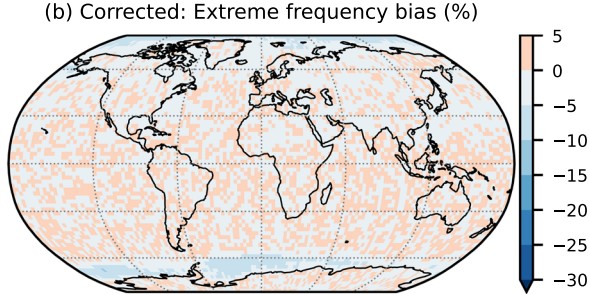

### (c) North Atlantic extreme frequency (mean bias: -33%)

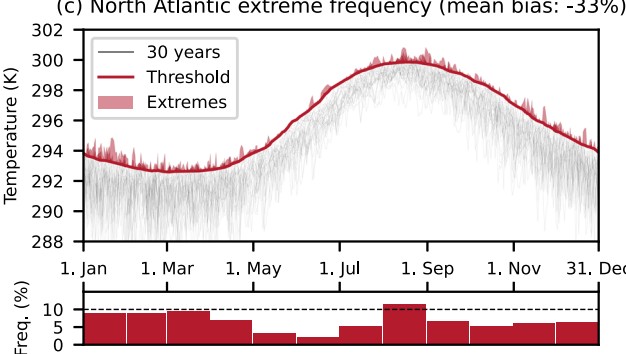

### (d) Corrected: North Atlantic extreme frequency (0%)

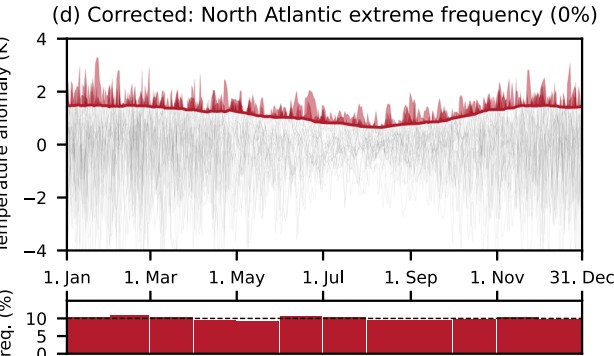

**Fig. 1 | Bias in the frequency of temperature extremes in ERA5. a** Spatial distribution in the frequency of daily maximum temperature extremes based on exceedances of the 90th percentile using a 31-day running window (TX90p31w) in the period 1961–1990. **c** Daily thresholds (thick red line), exceedances (red shading), and monthly averaged frequencies (bars) for a selected grid cell in the North Atlantic. **b, d** same as (**a, c**) but with the bias correction applied before the threshold calculation (discussed in the second part of the manuscript). The shown grid cell is marked with a red rectangle in **a**, additional grid cells are marked by orange rectangles and shown in Fig. S2 in the supplementary information.

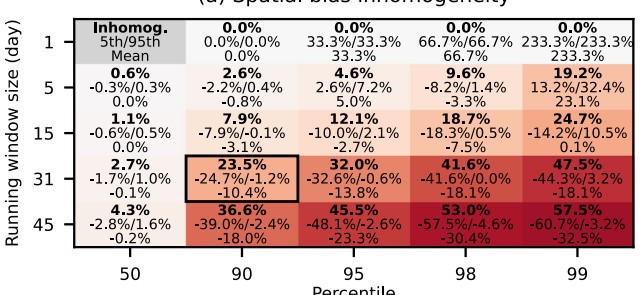

**Fig. 2 | Bias in the extreme frequency for different percentiles in synthetic data.** **a** Evolution of in-base (orange) and out-of-base (green) bias for two window sizes and different percentile values without seasonal cycle. **b** Difference between the out-of-base and in-base biases in **a**. **c** In-base bias without (orange) and for a moderate (blue) and strong (red) seasonal cycle. The 5-day running window is dashed, and the 31-day window is solid in all panels.

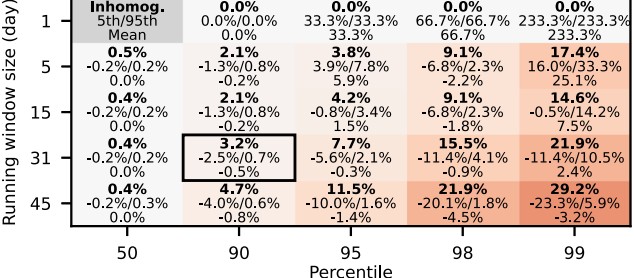

**Fig. 3 | Spatially aggregated bias statistics in ERA5. a** Statistics for different combinations of percentile and window size, with the highlighted rectangle showing the case corresponding to Fig. 1a (see Fig. S4 for additional bias maps). Spatial inhomogeneities (bold value in the top line) are defined as the difference between regions with high and low bias and provide a measure of the unequal treatment of regions with different biases. The low and high biases, defined as the spatial 5th and 95th percentiles, respectively, are shown in the middle line. The global mean bias is shown in the bottom line. The shading is locked to the inhomogeneities, with higher values shown in darker red. **b** Same as **a** but with the bias correction applied before the threshold calculation.

Crucially, however, this advantage of the longer window does not hold in the presence of even a moderately strong seasonal cycle (equivalent to the median in ERA5) as shown in Fig. 2c. While the bias for the 5-day window hardly changes compared to the case without a seasonal cycle, the 31-day running window leads to heavily increased bias. In fact, the running window bias introduced by the seasonally running window in combination with a seasonal cycle is larger than the bias due to the limited number of samples in Fig. 2a and larger than the difference in Fig. 2b for all but the highest percentiles. The evolution of this running window bias across the seasonal cycle is shown in Fig. S3 for a range of different window sizes.

**Pitfalls in the interpretation of temperature extremes**
The presence of the running window bias undermines the main assumption about percentile-based extreme definitions made (more or less explicitly) by many studies, namely that they allow extremes equally across the seasonal cycle [e.g.,[4,9,14]]. In the following, we discuss three additional pitfalls in the interpretation of extremes and connect them to further assumptions about percentile-based extremes made in the literature: (1) spatial inhomogeneities, (2) artificial dataset differences, and (3) spurious change signals.

(1) An important argument for the use of percentile-based indices is that they are based on thresholds following the local temperature distribution, thus making extremes and their changes comparable between regions [e.g.,[15,30]]. However, for a fair comparison, the probability for extremes should be the same across regions, which is undermined by the spatial variations of the bias (see Fig. 1a). As a measure for the unequal treatment of regions with low and high bias in a comparison we define spatial inhomogeneity as the difference

between the spatial 5th and 95th percentiles of grid cell biases. Figure 3 shows the spatial inhomogeneities for different combinations of percentiles and window sizes. They generally worsen with increasing percentile and window size, for TX90p31w, the inhomogeneity reaches ~25%. Note that for high percentiles and small window sizes, the mean bias greatly increases due to a lack of samples, so these cases are also problematic even though they have little or no spatial inhomogeneities (see ref. 8 for a detailed discussion). For TX90p5w (recommended by the ETCCDI) the global mean bias is almost zero, which is also reflected in a small inhomogeneity of ~3%, showing that this setup allows a fair comparison between regions. We note, however, that for such short window sizes the limited number of samples also leads to a bias (Fig. 2a) and to strong day-to-day variations in the extreme threshold, which might also pose a problem. Finally, we also show the 50th percentile in Fig. 3 to demonstrate that central estimates are not affected by the running window bias.

(2) Another important argument for percentile-based indices is that they provide an implicit bias correction of models compared to observations [e.g., refs. 2, 6]. The underlying extreme frequencies are, hence, assumed to be the same when analysing derived heatwave properties, such as duration, area, or cumulative heat. This means that any model-observation differences in these derived metrics are interpreted as higher-order differences. However, differences in these metrics can also result from a diverging bias structure leading to artificial dataset differences. Figure 4a shows the CMIP6 multi-model mean artificial difference to ERA5 and reveals several regions where model-observation differences exceed ± 10% robustly across most of the 26 models used. To illustrate the extent of possible differences, we compare a grid cell in the Amazon region between ERA5 and the

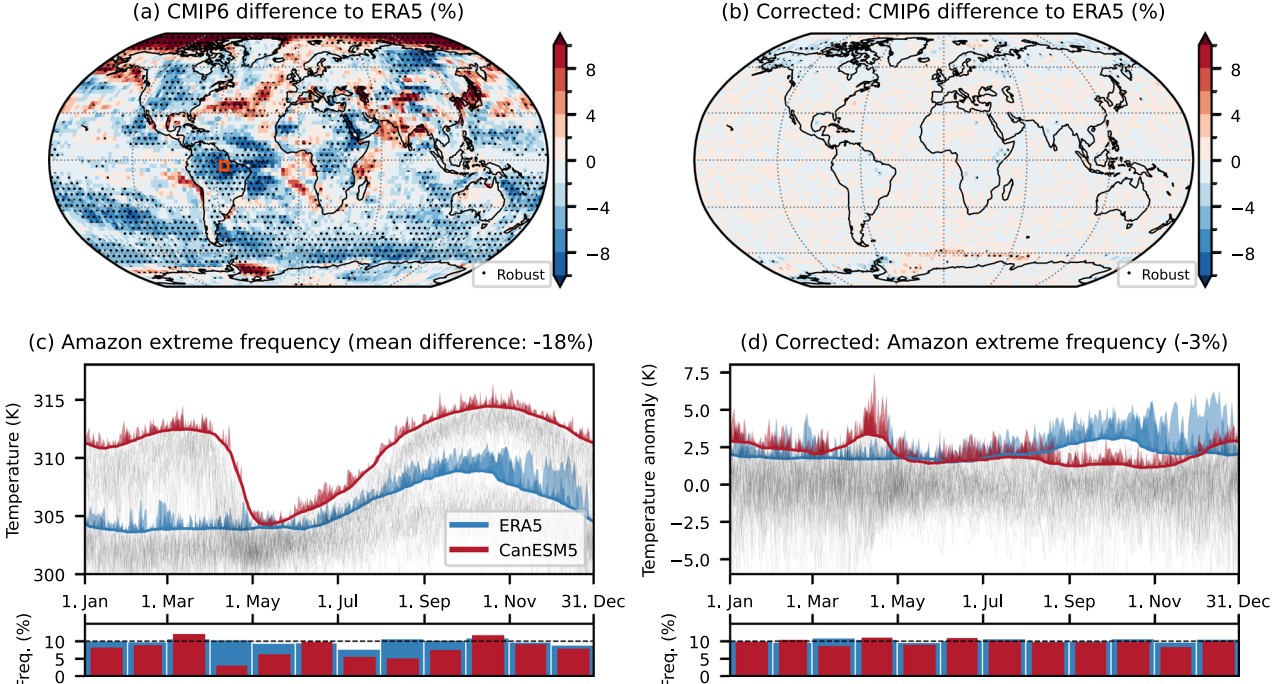

**Fig. 4 | Artificial frequency differences between CMIP6 and ERA5. a** CMIP6 multi-model mean extreme temperature (TX90p31w) frequency relative to ERA5 in the period 1961–1990. **c** CanESM5 extreme frequency for a selected grid cell with large differences in the Amazon region compared to ERA5. **b**, **d** same as **a**, **c** but with the bias correction applied before the threshold calculation. The shown grid cell is marked with a red rectangle in **a**. Areas with more than 80% of CMIP6 models agreeing on the sign and differences larger than ± 2% (lightest shading) are stippled.

CanESM5 model in Fig. 4c. The differences in the extreme frequency exceed 70% in April, due to a wrong representation of the mean seasonal cycle in CanESM5. Such a difference could easily be misinterpreted as a wrong representation of extremes in CanESM5 compared to ERA5 but is, in fact, founded in a wrong mean seasonal cycle interacting with the running window.

(3) A final argument for percentile-based temperature extreme definitions is that they account for the warming trend when using shifting baselines [e.g., refs. 20, 21]. This is intended to allow, for example, the investigation of non-linear changes in heatwave properties under climate change and, generally, follows an interpretation of extremes as events, which are rare by definition, even in a warming world [e.g., refs. 31, 32]. However, the running window bias can shift between time periods and translate into spurious change signals. Figure 5a shows such spurious change signals as the difference between two periods, for which separate thresholds are calculated: historical (1961–1990) and future (2071–2100) using the high emission scenario SSP3-7.0. Figure 5d shows a grid cell in the Arabian Sea for the CanESM5 model as an example. The decrease in the amplitude of the mean seasonal cycle in the future leads to a greatly reduced running window bias and, therefore, a striking increase in extreme day frequency by >30% in the annual mean. For the month of September, where extremes were basically absent in the historical period, the frequency even increases by a factor of 15.

**A simple solution to eliminate the running window bias**

To essentially eliminate the running bias even when using a 31-day window, we suggest to account for the seasonal variations first, to avoid mixing them into the extreme threshold. A simple way of doing so is to remove the mean seasonal cycle before calculating the threshold and subsequent extremes (see methods for details). In fact, this has been proposed already as early as 1999 by Folland et al.[33] and Jones et al.[34] and later been reiterated by Zhang et al.[8], but it has not been applied in the recent literature to the best of our knowledge. One

reason for this may be that these earlier studies did not discuss the potential for biased results due to the seasonal cycle we show here.

Figure 1b, d clearly reveals the benefit of removing the mean seasonal cycle before calculating extreme frequencies. The global mean frequency bias is reduced from −10% to −0.5% and from −33% to 0% for the example grid cell in the North Atlantic (see Fig. S2 for the other example grid cells). As a result, the spatial inhomogeneity is also vastly reduced from 24% to 3% (Fig. 3), and hardly any systematic bias pattern remains.

Exceptions are regions in the Southern Ocean and Arctic Ocean where the correction leads to a slight increase in bias. In the Southern Ocean the upper end of the temperature distribution is mostly constant throughout the year, basically following the 0 °C isotherm (Fig. S2i) when the seasonal cycle is still included. In the Arctic Ocean the same is true during northern hemispheric summer (Fig. S2g). At the same time, these regions also exhibit a strong seasonality in the day-to-day variability, with the amplitude being considerably lower during summer. This leads to an increase in the seasonality of the 90th percentile threshold when the mean seasonal cycle is removed and, hence, to a slight increase in the running window bias. However, the bias in these regions is small overall, with the highest amplitudes staying below 10% in both the uncorrected and corrected cases.

Artificial dataset differences are also greatly reduced when the mean seasonal cycle is first removed (Fig. 4b). The fraction of grid cells with robust differences (stippling in Fig. 4a,c) decreases from 37% to 2% when the correction is applied. Figure 4d shows the example of the grid cell in the Amazon that removes the mean seasonal cycle before calculating the threshold accounts for offsets between the datasets. The same holds true for differences between time periods, when using multiple base periods as shown in Fig. 5. The spurious change signal showcased for the grid cell in the Arabian Sea is essentially eliminated, and the only robust differences remaining are at high latitudes, connected to the changes in day-to-day variability discussed above.

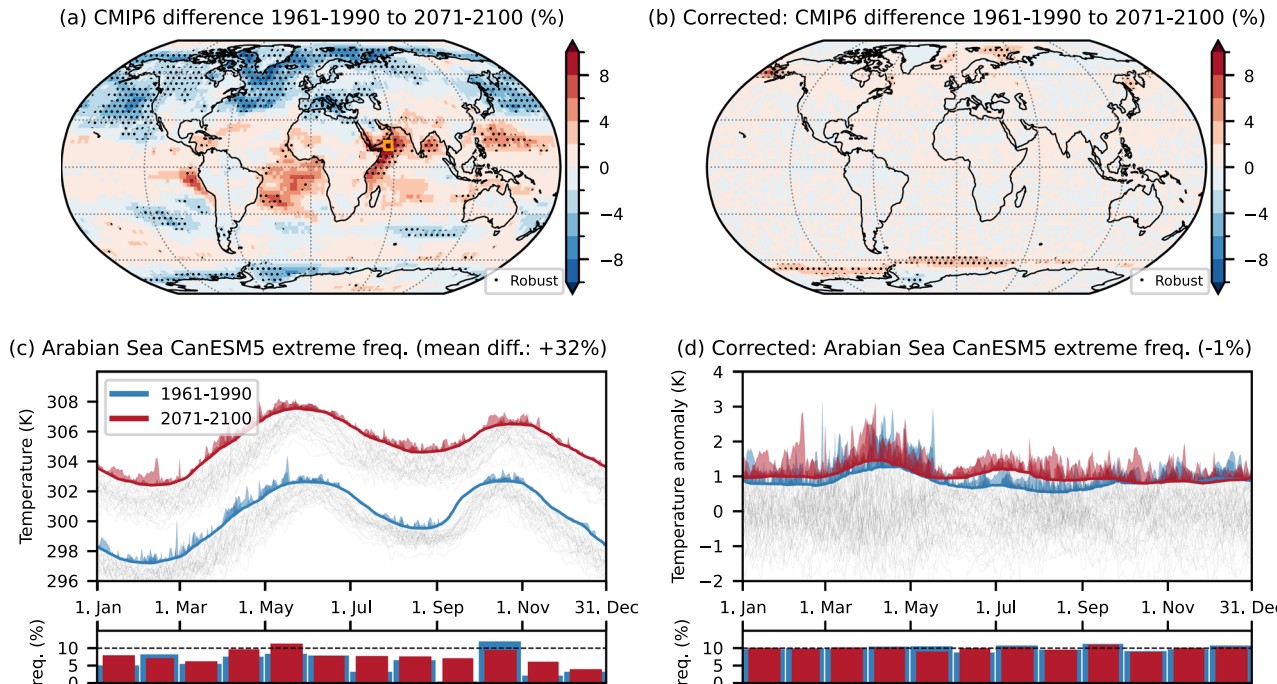

**Fig. 5 | Spurious change signals between 1961 and 1990 and 2071 and 2100 in CMIP6. a** CMIP6 multi-model mean difference between the periods 1961–1990 and 2071–2100 using separate extreme thresholds for each period. **c** CanESM5 extreme frequency in both periods for a selected grid cell with a strong signal in the Arabian Sea. **b**, **d** Same as **a**, **c** but with the bias correction applied before the threshold calculation. The shown grid cell is marked with an orange rectangle in **a**. Areas with more than 80% of CMIP6 models agreeing on the sign and differences larger than ±2% (lightest shading) are stippled.

## Impact of the bias on heatwave changes

Now we use the corrected extreme frequencies as a reference to show the impact of the running window bias on climate change signals. We focus on two metrics: (1) annual mean extreme frequency, which corresponds to the rest of our study and (2) summer heatwaves. The second metric is calculated from the extreme frequencies by restricting them to the respective extended summer season in both hemispheres and then applying a 3-day persistence criterion following, for example, Lyon et al.[16] and Perkins–Kirkpatrick and Lewis[17]. As change metric we show the CMIP6 multi-model mean ratio between 2071–2100 and 1961–1990 frequencies following Fischer and Schär[4] (see methods for details). Extremes in the future are calculated using a fixed 1961–1990 threshold in this section.

For the change in the extreme frequency, we can set an upper limit to the ratio based on theoretical considerations: if we expect a historical extreme frequency of about 10%, then a tenfold increase should be the maximum, corresponding to every day being extreme in the future. However, the historical extreme frequency can be considerably lower due to the running window bias, as we have shown. At the same time the extreme frequency greatly increases in the future when using a fixed baseline, leading to a decrease in the bias everywhere on the globe (Fig. S6). Figure 6 (top row) shows changes exceeding a factor of 13 for the biased case, while our correction leads to increases being mostly limited to the theoretical maximum. A direct comparison of extreme changes between biased and corrected cases is shown in Fig. 6c and reveals that the bias leads to an overestimation of extreme changes in most regions. The overall pattern is similar to the historical bias shown in Fig. 1a, which consists of our expectations based on the general decrease of bias in the future.

Finally, we repeat the same steps as above for the case of extended summer heatwaves, which is a frequently used metric in more impact-focused studies. For this case, no a priori theoretical maximum for the increase exists since the 3-day persistence criterion leads to spatially varying heatwave frequencies in the historical period (ranging between 3% to 7%; see Fig. S7a, d). Figure 6d shows the resulting bias in the heatwave ratio with ocean grid cells masked to highlight changes over land. The Arabian Peninsula is a particular hot spot of over-estimation which can exceed 30% here. This is founded in a strong historical bias combined with a strong reduction in bias in the future as heatwave frequency approaches 100% in this region (Fig. S6).

## Discussion

We have demonstrated that the percentile-based temperature extreme frequency can exhibit considerable deviations from the expected frequency due to a bias introduced by the use of running seasonal windows. This running window bias can vary with season, region, time period, and dataset undermining generally accepted properties of percentile-based extreme definitions. The bias changes in the future and, therefore, has implications also for climate changes studies. We also note that, while we have demonstrated the effect on hot extremes here, the same considerations apply equally to cold extremes.

One way to avoid the running bias is to limit the size of the running window to 5 days as recommended by the Expert Team on Climate Change Detection and Indices (ETCCDI), since such a window length causes only a negligible running window bias. However, the limited sample size in this case can also lead to a bias and to the sampling of day-to-day variability into the threshold, which may also pose a problem.

To allow the use of longer-running windows without bias, we recommend removing the mean seasonal cycle before calculating the extreme threshold. This essentially eliminates the bias as shown for the annual extreme frequency. Drawing on the corrected frequencies as reference, we show that the running window bias also affects summer heatwaves based on biased extreme frequencies. The effect on other derived extreme metrics and heatwave properties remains to be investigated.

To conclude, we strongly warn against the use of long-running windows without correction when calculating extreme thresholds. The

(a) Extreme frequency change (ratio)
2071-2100 relative to 1961-1990

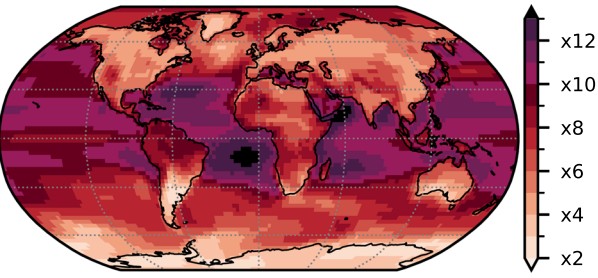

(b) Corrected: Extreme frequency change (ratio)
2071-2100 relative to 1961-1990

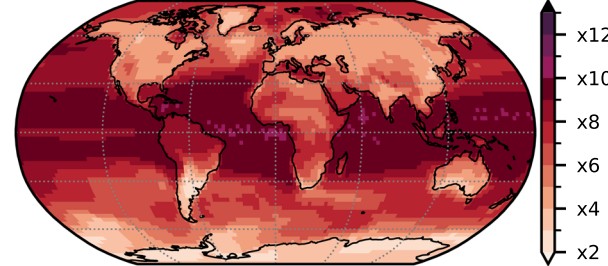

(c) Extreme frequency change bias (%)

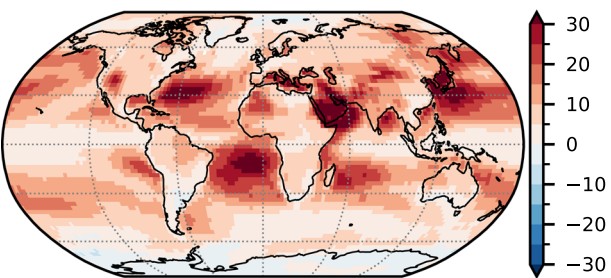

(d) Heatwave frequency change bias (%)
Extended summer land surface

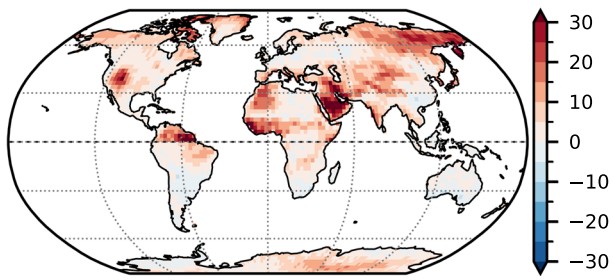

**Fig. 6 | Change in the extreme frequency between 1961 and 1990 and 2071 and 2100 in CMIP6.** Top row: Change in the frequency of daily maximum temperature extremes as the ratio of the 2071–2100 and 1961–1990 periods, using 1961–1990 as fixed baseline for the **a** uncorrected and **b** corrected cases. Ratios above ×10 are highlighted in purple as they are only possible due to the bias in the historical period. **c** Bias in the extreme frequency change relative to the corrected case. **d** Same as **c** but restricted to the extended summer season and with an additional 3-day persistence criterion. To highlight changes on land, ocean grid cells are masked for this case. Intermediate steps for **d** are shown in Fig. S7.

use of such a biased method is never advisable, even though the impacts on derived metrics might not always be strong or immediately apparent. Many studies, naturally, adopt methods from previous work, so avoiding the use of biased methods also helps to prevent their spread. Crucially, the same biased method may have a negligible effect in one setting and a large impact in another.

## Methods
### ERA5 data
We draw on data from the fifth generation of the European Centre for Medium-Range Weather Forecasts (ECMWF) Retrospective Analysis (ERA5[35];). We use hourly instantaneous 2 m surface air temperature with a native resolution of 0.25° × 0.25°, re-sampled to the daily maximum value and a spatial resolution of 2.5° × 2.5° using conservative remapping and restricted to the 30-year period 1961–1990. To simplify seasonal cycle-related calculations, we exclude February 29th from leap years so that each of the 30 years has 365 days.

### CMIP6 data
To complement ERA5, we also use daily maximum 2 m surface air temperature from 26 models from the Coupled Model Intercomparison Project Phase 6 (CMIP6[36];) as listed in table S1 in the supplementary information. These models represent an "ensemble of opportunity" of all models that are available from the ETH Zurich CMIP6 next-generation archive[37] and provide daily maximum temperature for the historical period and the Shared Socio-economic Pathway 3-7.0 (SSP3-7.0[38];) in the future. While we acknowledge that this pool of models does not constitute a representative or exhausting sample of the full diversity of models in the CMIP6 archive [e.g., refs. 39, 40] our aim here is mainly to showcase the potential for biases in the calculation of extremes rather than an in-depth model evaluation and we, therefore, chose this pragmatic approach. The pre-processing applied to the CMIP6 models is identical to ERA5, except that we use the period 2071–2100 in addition.

### Calculation of percentile-based extreme thresholds
Extreme thresholds are defined for each dataset and grid cell separately by calculating the $p$-th percentile over 30 years and the $\pm d$ days surrounding each day of the year. Temperature extremes for a given day are then defined as exceedances of the respective percentile value centred on the same day of the year. In our study, we focus on the 90th percentile calculated using a 31-day window (termed TX90p31w) but also investigate a range of other combinations of percentiles and window sizes to identify their interaction with the seasonal cycle.

The calculation of percentiles is done empirically on the pooled data from the 30 years and $\pm d$ days, so on $30 \times 31 = 930$ values for TX90p31w. Modern programming languages offer a range of choices to calculate the percentile and the choice of method can influence the results, in particular for high percentiles, which are exceeded by only a few values. For the main part of the manuscript we use the method "linear", which is the default setting in Python's NumPy package (https://numpy.org/doc/stable/reference/generated/numpy.percentile.html), in R (https://stat.ethz.ch/R-manual/R-devel/library/stats/html/quantile.html), and in Julia (https://docs.julialang.org/en/v1/stdlib/Statistics/). It can, therefore, be assumed to be a frequently employed setting for the calculation of percentile-based thresholds. For our comparison to the results from Zhang et al.[8] in Fig. 2 we, instead, use NumPy's "weibull" method to be consistent with their approach. Biases similar to Fig. 2 but using "linear" are shown in Fig. S8 in the supplementary information.

### Calculation of extreme frequency, bias, and difference
The extreme frequency $f(p, w)$ is the fraction of days exceeding the extreme threshold in a given month or the entire year in percent:

$$f(p,w) = \frac{x_{\text{exceed}}(p,w)}{x_{\text{base}}} \times 100\%, \tag{1}$$

with $p$ indicating the percentile and $w$ the window size.

The running window bias $f'(p,w)$ is defined as the extreme frequency minus the expected extreme frequency $f_{exp}(p) = 100 - p$ (so 10% for the 90th percentile, as an example) divided by the expected frequency in percent:

$$f'(p,w) = \frac{f(p,w) - f_{exp}(p)}{f_{exp}(p)} \times 100\% \qquad (2)$$

In Fig. 6 we use the corrected frequency as $f_{exp}$.

Equivalently, the difference $\Delta f$ between two datasets or periods is calculated relative to the reference extreme frequency $f_{ref}(p, w)$ following:

$$\Delta f(p,w) = \frac{f(p,w) - f_{ref}(p,w)}{f_{ref}(p,w)} \times 100\% \qquad (3)$$

With the reference being ERA5 in Fig. 4 and the historical period in Fig. 5.

### Removing the mean seasonal cycle

The mean seasonal cycle is calculated over 30 years on a day-of-the-year basis (and, hence, without any running window) for each grid cell separately. We choose this approach here for its simplicity, an alternative option would be to use a running window in the calculation for the mean seasonal cycle to increase the sample size. This is not problematic as central estimates are not affected by the running window bias (see Fig. 3).

For the historical 1961–1990 period the mean seasonal cycle is always calculated in the same period. For the future two options exist: (1) for Fig. 5 the mean seasonal cycle for the 2071–2100 period is used, mirroring the percentile calculation with multiple baselines; (2) for Fig. 6 the historical 1961–1990 seasonal cycle is used.

### Calculation of summer heatwaves and climate change signals

In a first step extremes are constrained to the extended summer season: May-September in the northern and November-March in the southern hemisphere. Then heatwaves are defined at a grid cell basis as at least three consecutive extreme days. Finally, to select land grid cells we apply a common land-sea mask to the regridded data, based on Natural Earth (https://www.naturalearthdata.com/) and implemented in Python's regionmask package (https://regionmask.readthedocs.io).

Climate change signals between the periods 1961–1990 and 2071–2100 (using the first period as fixed baseline for both cases) in the last section are calculated as extreme frequency ratio $r$ following Fischer and Schär[4]:

$$r = \frac{f_{2071\text{-}2100}}{f_{1961\text{-}1990}} \qquad (4)$$

For such a comparison of frequencies between in-base and out-of-base periods (Fig. 6) Zhang et al.[8] recommend the use of a cross-validation approach to avoid an artificial frequency jump between the periods. Their cross-validated extreme frequency is calculated as mean over 29 folds for each of the in-base years (see ref. 8 for details). This means the output is no longer a time series of individual days exceeding the threshold or not, but an average extreme frequency. This prohibits the calculation of heatwaves from the extreme frequency by applying the 3-day persistence criterion. Therefore, we, here, directly use the extreme frequency to calculate heatwaves in the base-period as well as outside, again, following Fischer and Schär[4]. In this context, we note that the jump for the 90th percentile using a 31-day window is very small at only ~3% as can be seen from Fig. S8. In addition, we focus on a comparison between the uncorrected and corrected frequencies in Fig. 6, which are both affected equally by the jump.

### Creation of synthetic time series

For the creation of the synthetic time series, we combine auto-correlated white noise emulating day-to-day variability and a sine-function emulating the seasonal cycle. The white noise is defined to have a mean of zero, a standard deviation of one, and a lag 1-day auto-correlation of 0.8. The lag 1-day auto-correlation of 0.8 is used to be consistent with the work by Zhang et al.[8] and because it is reasonably close to the median lag 1-day auto-correlation of the de-seasonalized day-to-day variability in ERA5, which is 0.76 in the period 1961–1990.

The amplitude of the sine function is set to three distinct values so that the ratio of the standard deviation over the seasonal cycle and the standard deviation over the white noise (which is one by definition) represents three cases: 0 (no seasonal cycle), 1.8 (same as the spatial median in ERA5), and 3 (corresponding to the 90th percentile in ERA5).

The ratio of standard deviations $\Delta s$ in ERA5 is calculated for each grid cell separately following:

$$\Delta s = \frac{<\overline{x}(d)>}{<x(y,d) - \overline{x}(d)>}, \qquad (5)$$

where the enumerator is the standard deviation of the mean seasonal cycle (calculated on a day-of-the-year basis with a window size of 1 and the denominator is the mean over standard deviations calculated from the de-seasonalized day-to-day variability for each year separately.

Following Zhang et al.[8] we produce a synthetic 30-year interval and calculate the percentile threshold, exceedance, and bias from it (in-base case). For the out-of-base case we produce a separate 30-year interval which is evaluated against the threshold calculated from the first 30 years. This process is repeated 5000 times, and the presented biases are the average over all iterations.

## Data availability

All raw data used in this study are freely available for research applications. ERA5 hourly 2 m temperature: https://cds.climate.copernicus.eu/cdsapp#!/dataset/reanalysis-era5-single-levels?tab=overview; CMIP6 daily maximum 2 m surface air temperature: https://esgf-node.llnl.gov/; An example dataset providing extreme frequencies for ERA5 using a 31-day window for the 90th percentile is available here: https://doi.org/10.5281/zenodo.10639317.

## Code availability

The code to re-create the figures in the main manuscript is available at: https://github.com/lukasbrunner/running_window_bias.

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

## Acknowledgements

The authors thank Erich M. Fischer and Gabriele C. Hegerl for helpful discussions on the manuscript, and Urs Beyerle for downloading the CMIP6 data for the ETH next-generation archive.

## Author contributions

This work was conceptualised and written by L.B. with contributions from A.V. Data acquisition, analysis, and visualisation by L.B.

## Competing interests
The authors declare no competing interests.
