## [Peer Review File · Nature Communications]

Pitfalls in diagnosing temperature extremesREVIEWER COMMENTS

Reviewer #1 (Remarks to the Author):

This paper shows that the running window used to compute calendar-day percentiles introduces biases in the estimation of temperature extremes defined relative to these percentiles. The proposed solution is to remove the seasonal cycle before computing the percentiles and extremes. I found this paper to be convincing, with results that are significant and important for the field. A few minor questions/suggestions are listed below.

Line 39 – to be a little clearer, maybe say “150 values per calendar day for the percentile calculation”

Figure 1 – why is the ocean masked out in 1b but not 1a?

Line 66 – What is the significance of these two grid cells? Are these the most extreme examples?

Figure A4 – does the described phenomenon from the Southern Ocean apply to points in the Arctic as well? Either way, I think this caveat is worth mentioning in the main text around line 153.

I recognize this is not the focus of this paper, but I am just curious if you have considered whether this issue and the resulting recommendation apply to precipitation percentiles and indices

Reviewer #2 (Remarks to the Author):

NCOMMS-23-45841-T

Pitfalls in diagnosing temperature extremes

Lukas Brunner and Aiko Voigt

This manuscript investigates the effect of the running window length on the frequency of exceedance of percentile based indices (i.e. Tx90p). The manuscript is interesting and raises an important question in the methodology of essentially any work investigating the effect of climate change on e.g. heat extreme indices. However, I have some doubts on the real practical implication of this issue when applied to e.g. climate change application (which is also partly acknowledged by the authors themselves).

I fully understand the issue is indeed relevant especially for monitoring applications (e.g. early warning systems) but I am not fully convinced that the issue reported by the authors (and their solution, which by the way I appreciate in its simplicity) would really matter when used in climate change context (especially if compared to e.g. other kind of uncertainties).

Specific comments are listed below:

Figure 1: A) It is a bit hard to distinguish the different levels of bias in panels a) and b) because of the monochromatic colour scale, especially when printed (on screen it is indeed a bit better). B) Similarly, the two selected grid points are not immediately evident. Please mark them more clearly. C) the numbers in the bottom of the frequency bias bars (panels c and d) are hard to read on both online and printed versions. D) It is striking to me that over most of the land regions (Europe, Australia, sub-equatorial Africa, etc.) the bias is actually relatively low (<5%? Or even positive). Do you have an explanation for that? E) for comparison, it would be interesting to see also maps of bias when a 5day window is used. Given my point above (D) it would be important to check indeed if the bias is dramatically different when a 31d window is used instead of a 5d.

L41: Please note that The Heat Wave Magnitude Index (HWMI, Russo et al 2104) has been (extensively) modified and most of the quoted paper use the Heat Wave Magnitude Index-daily (HWMId) defined by Russo et al 2015.

L51 typo: a simple

L130-141: I assume that the Tx90p index in 2071-2100 is calculated using the same 2071-2100 period as reference (which is implicit, I guess, but not explicitly mentioned). However, in any study of climate change, the reference period used would be the same for the past and the future (e.g. 1961-1990). I wonder again how much of a difference the length of the window would make on this difference (i.e. Tx90p in the future compared to that in the present using the same reference period). Another interesting comparison would be to compare this difference on a shorter time periods, e.g. 1991-2020 vs 1961-1990 (again using the same reference period) based on ERA5 data.

Figure A4: A) As you show the results with and without seasonal cycle for one point (c,d) you should do the same for the other one (which is Figure 1c if I understand correctly?). B) Also, it would be interesting to have the same figure for the summer season land point only data, as in figure 1b (and corresponding results for the point in the Arabian peninsula figure 1d).

L168-184 Here to me is the main issue with this work: “he WSDIs of the strongest summer heatwaves discussed in Russo et al. [29] are only weakly affected by the bias as they are typically centred in high summer when the running window bias is small or even positive. In addition, the strongest heatwaves often exceed the threshold by a large margin, making them less sensitive to the running window bias.” It seems to me that, in any practical terms, especially for studies of the impact of climate change on e.g. heatwave characteristics defined by e.g. the HWMId index, the real effect of the window length is (very) limited. It would be interesting if the authors would show, similar to their figure 1) maps of .g. WSDI both in the present and in the far future (based on the same 1961-1990 reference period) and discuss the impact of the window length (5d vs 31d) on these results, which are those typically discussed in most of the works quoted in the literature (and which the author implicitly assume, in the introduction, would be the most affected by the window length issue). Obviously, the analysis should be conducted with and without the seasonal cycle, to show the advantage of the simple solution proposed here.

Reviewer #3 (Remarks to the Author):

This is a nice paper showing that using sliding windows to define percentile-based temperature extreme thresholds introduces bias into statistics of extreme event frequency due to the presence of the seasonal cycle. The paper is well written with nice figures that clearly show the result.

While I think this is a useful and interesting paper for researchers in the heat extremes field, I am not sure that it is appropriate for Nature Comms. I think it would likely be better suited to a field-specific journal like GRL or ERL (or similar).

I have a few comments that I think the authors could consider to improve the paper:

1. The final section introduces a modified methodology to eliminate the sliding window bias, but the results showing this method are in an appendix. I suggest bringing these to the main manuscript and put them side by side the biased results so a reader can easily see the difference!
2. I think the authors could broaden the reach of this paper by including a full analysis of extreme heat changes using the new method and comparing them to past results. Right now this is a methods paper, which is great, but it doesn't really have any scientific results about heat. You could analyze CMIP6, show the differences/biases in model seasonal cycle, and show estimated changes in extreme heat frequency under global warming scenarios.

Reviewer #1 (Remarks to the Author):

This paper shows that the running window used to compute calendar-day percentiles introduces biases in the estimation of temperature extremes defined relative to these percentiles. The proposed solution is to remove the seasonal cycle before computing the percentiles and extremes. I found this paper to be convincing, with results that are significant and important for the field. A few minor questions/suggestions are listed below.

Thank you to the reviewer for the positive evaluation of our paper. Please find our answers to the comments in bold below.

For your convenience, we also summarize the main changes in the manuscript based on the other reviewers' comments here:

- **We now show the corrected frequencies side-by-side with the biased results in figures 1, 3, 4, and 5.**
- **We have added a section addressing the impact of the bias on heatwave changes**
- **We have added a discussion section at the end**

Line 39 – to be a little clearer, maybe say “150 values per calendar day for the percentile calculation”

Done, thank you.

Figure 1 – why is the ocean masked out in 1b but not 1a?

Figure 1b no longer exists in the revised manuscript. We now show a similar ocean-masked case in figure 6. This is done to highlight the bias for land areas which are often the focus of more impact-focused studies. We have made our reasoning more clear in the revised manuscript and also show a version of figure 6 without ocean mask in figure S7 in the supplement.

Line 66 – What is the significance of these two grid cells? Are these the most extreme examples?

In the revised manuscript we now show and discuss one example grid cell in the main paper and additional grid cells in the supplement.

The grid cell shown in the main manuscript represents an area with a strong bias (-33%) but not the most extreme case (which would be <-50% for a 31-day window). It is mainly intended to showcase the effect of the running window bias along the seasonal cycle.

Figure A4 – does the described phenomenon from the Southern Ocean apply to points in the Arctic as well? Either way, I think this caveat is worth mentioning in the main text around line 153.

Thank you to the reviewer for raising this question. Indeed, the same reasoning applies also to the Arctic. We now discuss this caveat in more detail in the text (line 154 in the revised manuscript):

Exceptions are regions in the Southern Ocean and Arctic Ocean where the correction leads to a slight increase in bias. In the Southern Ocean the upper end of the temperature distribution is mostly constant throughout the year, basically following

the 0°C isotherm (figure S2i) when the seasonal cycle is still included. In the Arctic Ocean the same is true during northern hemispheric summer (figure S2g). At the same time, these regions also exhibit a strong seasonality in the day-to-day variability, with the amplitude being considerably lower during summer. This leads to an increase in the seasonality of the 90th percentile threshold when the mean seasonal cycle is removed and, hence, to a slight increase in the running window bias. However, the bias in these regions is small overall, with the highest amplitudes staying below 10 % in both the uncorrected and corrected cases.

I recognize this is not the focus of this paper, but I am just curious if you have considered whether this issue and the resulting recommendation apply to precipitation percentiles and indices

From the theoretical understanding we build in our work, it seems clear that the running window bias will affect all percentile-based indices if the variable in question has a strong seasonal cycle amplitude compared to the amplitude of the residual day-to-day variability. While this may be the case also for precipitation in some regions we do not expect precipitation to be affected to the same degree as temperature due to its higher variability on short time scales.

Reviewer #2 (Remarks to the Author):

This manuscript investigates the effect of the running window length on the frequency of exceedance of percentile based indices (i.e. Tx90p). The manuscript is interesting and raises an important question in the methodology of essentially any work investigating the effect of climate change on e.g. heat extreme indices. However, I have some doubts on the real practical implication of this issue when applied to e.g. climate change application (which is also partly acknowledged by the authors themselves).

I fully understand the issue is indeed relevant especially for monitoring applications (e.g. early warning systems) but I am not fully convinced that the issue reported by the authors (and their solution, which by the way I appreciate in its simplicity) would really matter when used in climate change context (especially if compared to e.g. other kind of uncertainties).

Thank you to the reviewer for evaluating our manuscript and for the helpful comments. In the revised manuscript we better highlight the relevance of our findings and how they might affect the investigation of temperature extremes in various ways.

For this, we have restructured the section *Pitfalls in the interpretation of temperature extremes* and now more clearly point out how assumptions made in earlier studies are impacted by the bias we discuss.

In addition, we have added a section specifically addressing the impact of the bias on change studies (see our answer to the last comment) and a discussion section.

Specific comments are listed below:

Figure 1:

A) It is a bit hard to distinguish the different levels of bias in panels a) and b) because of the monochromatic colour scale, especially when printed (on screen it is indeed a bit better).

B) Similarly, the two selected grid points are not immediately evident. Please mark them more clearly.

C) the numbers in the bottom of the frequency bias bars (panels c and d) are hard to read on both online and printed versions.

Thank you to the reviewer for these suggestions. We have reduced the number of color levels in figure 1 to make them better distinguishable and improved the contrast for the highlighted grid cells. In addition, we have increased the overall font size for all figures.

D) It is striking to me that over most of the land regions (Europe, Australia, sub-equatorial Africa, etc.) the bias is actually relatively low (<5%? Or even positive). Do you have an explanation for that?

Thank you for raising this question. We now show several additional example grid cells in the supplement (figure S2) to cover the diversity of cases over ocean and land. We have also added some discussion on this point to the manuscript (line 68):

The strongest bias is, hence, found in regions and seasons with a strong seasonal gradient but weak day-to-day variability, as showcased in figure 1c and figure S2. While many of the strongest biases, therefore, occur over oceans there are also several land regions with considerable bias such as India and the western US (figure

S2a,c). Across most of Europe the strong seasonal cycle is offset by strong day-to-day variability and only a weak bias remains (figure S2e). In regions with weak seasonal variations, such as most of the tropics and the Southern Ocean (figure S2i) the bias is consequently negligible.

E) for comparison, it would be interesting to see also maps of bias when a 5day window is used.

We now show a range of different combinations of window lengths and percentile values in figures S4 and S5 for the biased and corrected cases, respectively. The mean bias for all cases is also shown in figure 3 in the main manuscript.

For a 5 day window the global mean running window bias is very small. However, other problems might arise when using such a short window as we discuss in line 111:

For TX90p5w (recommended by the ETCCDI) the global mean bias is almost zero which is also reflected in a small inhomogeneity of about 3 %, showing that this setup allows a fair comparison between regions. We note, however, that for such short window sizes the limited number of samples also leads to a bias (figure 2a) and to strong day-to-day variations in the extreme threshold, which might also pose a problem.

Given my point above (D) it would be important to check indeed if the bias is dramatically different when a 31d window is used instead of a 5d.

Bias values for different window sizes are now shown in figure S4 as mentioned above.

L41: Please note that The Heat Wave Magnitude Index (HWMI, Russo et al 2104) has been (extensively) modified and most of the quoted paper use the Heat Wave Magnitude Index-daily (HWMId) defined by Russo et al 2015.

Thank you for pointing this out, we have updated the citation.

L51 typo: a simple

Fixed, thank you.

L130-141: I assume that the Tx90p index in 2071-2100 is calculated using the same 2071-2100 period as reference (which is implicit, I guess, but not explicitly mentioned). However, in any study of climate change, the reference period used would be the same for the past and the future (e.g. 1961-1990). I wonder again how much of a difference the length of the window would make on this difference (i.e. Tx90p in the future compared to that in the present using the same reference period). Another interesting comparison would be to compare this difference on a shorter time periods, e.g. 1991-2020 vs 1961-1990 (again using the same reference period) based on ERA5 data.

We have adjusted the text and now explicitly mention that we use two reference periods. As the reviewer rightly points out, most studies apply a fixed baseline to investigate frequency changes under warming. However, there are also several (climate change) studies using shifting base periods as we discuss in the revised manuscript in line 131:

A final argument for percentile-based temperature extreme definitions is that they account for the warming trend when using shifting baselines [e.g., 31, 37]. This is intended to allow, for example, the investigation of non-linear changes in heatwave properties under climate change and, generally, follows an interpretation of extremes as events, which are rare by definition, even in a warming world [e.g., 2, 27]. However, the running window bias can shift between time periods and translate into spurious change signals. Figure 5a shows such spurious change signals as the difference between two periods, for which separate thresholds are calculated: historical (1961-1990) and future (2071-2100) using the high emission scenario SSP3-7.0. [...]

We have also calculated the bias in ERA5 for the period 1991-2020 (using the same period as reference) as well as the difference in bias compared to the period used in the manuscript (1961-1990). As can be seen in the figures below the overall bias pattern stays the same which is probably not surprising, given the closeness of the two periods.

Figure: (top) Extreme frequency bias in ERA5 in the period 1991-2020. (bottom) Relative difference between the periods 1961-1990 and 1990-2020, using two different thresholds calculated from the respective periods.

Figure A4:

A) As you show the results with and without seasonal cycle for one point (c,d) you should do the same for the other one (which is Figure 1c if I understand correctly?).

B) Also, it would be interesting to have the same figure for the summer season land point only data, as in figure 1b (and corresponding results for the pint in the Arabian peninsula figure 1d).

Thank you to the reviewer for raising this. We now always show the biased and corrected versions side by side (figures 1, 3, 4, and 5 as well as figure S2).

L168-184 Here to me is the main issue with this work: “he WSDIs of the strongest summer heatwaves discussed in Russo et al. [29] are only weakly affected by the bias as they are typically centred in high summer when the running window bias is small or even positive. In addition, the strongest heatwaves often exceed the threshold by a large margin, making them less sensitive to the running window bias.”

It seems to me that, in any practical terms, especially for studies of the impact of climate change on e.g. heatwave characteristics defined by e.g. the HWMId index, the real effect of the window length is (very) limited. It would be interesting if the authors would show, similar to their figure 1) maps of .g. WSDI both in the present and in the far future (based on the same 1961-1990 reference period) and discuss the impact of the window length (5d vs 31d) on these results, which are those typically discussed in most of the works quoted in the literature (and which the author implicitly assume, in the introduction, would be the most affected by the window length issue). Obviously, the analysis should be conducted with and without the seasonal cycle, to show the advantage of the simple solution proposed here.

Thank you to the reviewer for this comment. Showing the impact of the bias on heatwave changes with climate change is, indeed, a very relevant point to cover. In the revised manuscript we have added a new section addressing this question: *Impact of the bias on heatwave changes* (line 171).

Drawing on a similar setup as an earlier study by Fischer and Schär (2010), we use our correction to show that future heatwave frequency changes are overestimated by as much 30%. We have summarised the results in a new figure 6 in the revised manuscript and extended figures S6 and S7 in the supplement.

While we no longer refer to the WSDI directly, the metric we consider now is very similar also using a 3 day persistence criterion. In addition to heatwave frequencies throughout the year we now also look at frequencies constrained to the warm season which is the focus of many impact-focused studies.

Reviewer #3 (Remarks to the Author):

This is a nice paper showing that using sliding windows to define percentile-based temperature extreme thresholds introduces bias into statistics of extreme event frequency due to the presence of the seasonal cycle. The paper is well written with nice figures that clearly show the result.

Thank you to the reviewer for the positive evaluation of our paper and the helpful comments. Please find our answers in bold below.

While I think this is a useful and interesting paper for researchers in the heat extremes field, I am not sure that it is appropriate for Nature Comms. I think it would likely be better suited to a field-specific journal like GRL or ERL (or similar).

Since we are not sure if the reviewer is aware, we point out that we submitted this manuscript to the Nature Communications Special Issue on Weather and Climate Extremes (<https://www.nature.com/collections/cgdbbcfjii>), which we think is a good fit.

I have a few comments that I think the authors could consider to improve the paper:

1. The final section introduces a modified methodology to eliminate the sliding window bias, but the results showing this method are in an appendix. I suggest bringing these to the main manuscript and put them side by side the biased results so a reader can easily see the difference!

Thank you for this suggestion. We have restructured the figures and now always show the biased and corrected versions side-by-side (figures 1, 3, 4, 5)

2. I think the authors could broaden the reach of this paper by including a full analysis of extreme heat changes using the new method and comparing them to past results. Right now this is a methods paper, which is great, but it doesn't really have any scientific results about heat. You could analyze CMIP6, show the differences/biases in model seasonal cycle, and show estimated changes in extreme heat frequency under global warming scenarios.

Thank you to the reviewer for this suggestion. In the revised manuscript we now discuss the impact of the bias on changes in temperature extremes in a new section: *Impact of the bias on heatwave changes* (line 171).

For this, we draw on a similar setup as in an earlier study by Fischer and Schär (2010) and use our correction to show that future heatwave frequency changes are overestimated by as much 30%. We summarise the results in a new figure 6 in the revised manuscript and extended figures S6 and S7 in the supplement.

REVIEWERS' COMMENTS

Reviewer #2 (Remarks to the Author):

Thank you for addressing my comments and for the extended analysis which makes the paper more interesting and helpful for the community, in my opinion.

I think the manuscript can be published in this versions.